# Towards Robustness Prompt Tuning with Fully Test-Time Adaptation for CLIP's Zero-Shot Generalization

Ran Wang
Australian Artificial Intelligence Institute
FEIT, University of Technology Sydney
Sydney, Australia
ran.wang-2@student.uts.edu.au

Zhen Fang*
Australian Artificial Intelligence Institute
FEIT, University of Technology Sydney
Sydney, Australia
zhen.fang@uts.edu.au

Hua Zuo
Australian Artificial Intelligence Institute
FEIT, University of Technology Sydney
Sydney, Australia
hua.zuo@uts.edu.au

Jie Lu
Australian Artificial Intelligence Institute
FEIT, University of Technology Sydney
Sydney, Australia
jie.lu@uts.edu.au

## Abstract

In the field of Vision-Language Models (VLM), the Contrastive Language-Image Pretraining (CLIP) model has yielded outstanding performance on many downstream tasks through prompt tuning. By integrating image and text representations, CLIP exhibits zero-shot generalization capabilities on unseen data. However, when new categories and distribution shifts occur, the pretrained text embeddings in CLIP may not align well with unseen images, potentially leading to a decrease in CLIP's zero-shot generalization performance. To address this issue, many existing methods use test samples to update the CLIP model during testing through a process known as Test-Time Adaptation (TTA). Previous TTA techniques, such as image augmentation, can lead to overfitting given outlying samples, while methods based on teacher-student distillation can increase memory use. Further, these methods significantly increase inference time, which is a crucial factor in the testing phase. To improve robustness, mitigate overfitting, and reduce bias toward outlying samples, we propose a novel method: Self-Text Distillation with Conjugate Pseudo-labels (SCP), designed to enhance CLIP's zero-shot generalization. SCP uses gradient information from conjugate pseudo-labels to enhance the model's robustness toward distribution shifts. It also innovates by using a fixed prompt list to distil learnable prompts from within the same model, acting as a self-regulation mechanism that minimizes overfitting. Additionally, SCP is a fully test-time adaptation method that does not require retraining. It directly improves CLIP's zero-shot generalization at test time without increasing either memory overheads or inference time. In evaluations across three zero-shot generalization scenarios, SCP surpasses existing state-of-the-art methods in performance and significantly reduces inference time.

---

*Corresponding author

MM '24, October 28-November 1, 2024, Melbourne, VIC, Australia
© 2024 Copyright held by the owner/author(s).
ACM ISBN 979-8-4007-0686-8/24/10
https://doi.org/10.1145/3664647.3681213

## CCS Concepts

• **Computing methodologies** → **Transfer learning**.

## Keywords

Transfer Learning, Vision-Language Models, Test-Time Adaptation

**ACM Reference Format:**
Ran Wang, Hua Zuo, Zhen Fang, and Jie Lu. 2024. Towards Robustness Prompt Tuning with Fully Test-Time Adaptation for CLIP's Zero-Shot Generalization. In *Proceedings of the 32nd ACM International Conference on Multimedia (MM '24), October 28-November 1, 2024, Melbourne, VIC, Australia.* ACM, New York, NY, USA, 9 pages. https://doi.org/10.1145/3664647.3681213

## 1 Introduction

Within the field of large-scale vision-language models [11, 16], the Contrastive Language-Image Pre-training model (CLIP [29]) has demonstrated superior performance over traditional convolutional neural network models across a range of tasks, including few-shot classification and zero-shot generalization tasks [32]. The main idea of CLIP is an innovative integration of textual data with visual input through a dual-encoder framework. As such, CLIP consists of two encoders: an image encoder and a text encoder. It generates image features and text features to match a vast corpus of image-text pairs through a contrastive learning scheme. By this method, CLIP efficiently aligns visual and textual representations, allowing the model to interpret the visual content of images and to capture their associated textual representations. This distinctive strategy means CLIP is able to capture the intricate dynamics between visual and textual elements, significantly broadening its applicability and efficacy across diverse scenarios [46].

Recent research has integrated the concept of prompts into CLIP's text encoder. Taken from the domain of natural language processing (NLP [6]) this approach, known as prompt tuning, uses embedded prompts [21], such as *"a photo of a"* to let CLIP swiftly adapt to downstream tasks without the need to fine-tune the encoder. As a result, computation time is significantly reduced [34]. Further, to avoid the time-consuming task of manually crafting fixed prompts, analysts can use learnable prompts that dynamically change depending on the requirements of the task [49, 50].

This adaptability means CLIP can more effectively leverage its pre-trained knowledge, optimizing performance across a diverse range of tasks with minimal additional computational overhead.

However, while learnable prompt tuning enhances the capabilities of CLIP, this model still faces challenges with zero-shot generalization not only due to distribution shifts within unseen test data but also when confronted with entirely new labels or categories [44]. More specifically, images can contain drastically different content with huge variations in features. As an example, consider the differences between a sunny park and a snowy mountain landscape. Not only will the two images contain very different backgrounds, but unusual animal species and culturally-specific items will likely be encountered as well. These conditions represent both new category and new distribution shifts that can lead to a misalignment of image-text pairs. Overall, the result for CLIP will be diminished zero-shot generalization performance in new scenarios [42]. To address this challenge, previous methods have leveraged test data, using them to update the model during test-time through a process known as test-time adaptation (TTA).

While existing test-time adaptation methods can mitigate the effects of distribution shifts by using information from test data, these methods face challenges of their own. For example, most updates during test time depend on pseudo-labeling [24, 37], but pseudo-labeling can lead to a bias towards outlier samples when the distribution changes, diminishing the model's performance in changing environments. A more reliable method is image augmentation. This approach aims to minimize marginal entropy to achieve more certain results [34, 47]. However, its lack of adequate regularization can bias outcomes and lead to overfitting. Conversely, the teacher-student distillation method [8, 40] generally yields smoother results through distillation. The downside is that this strategy does increase memory use, particularly with large-scale vision-language models. It also substantially increases the inference time, both of which are critical considerations during testing.

To address these issues, we introduce a novel method: **S**elf text distillation with **C**onjugate **P**seudo-labels **(SCP)** for CLIP's zero-shot generalization. SCP employs conjugate pseudo-label to reduce the bias on outlier samples and self-text distillation to reduce overfitting. Specifically, instead of using only entropy information as is the case with traditional pseudo-labels, SCP uses conjugate pseudo-labels which are optimized based on both gradient and entropy information. This steers the model toward optimization, while reducing any bias toward outlying samples in the face of distribution shifts. The overall result is increased robustness. Further, to reduce overfitting, SCP implements self-text distillation as a form of self-regularization. It uses teacher-student distillation within the same frozen CLIP text encoder, which uses less memory than loading two separate CLIP models. The learnable prompt mechanism acts as the student model, and a fixed prompt list serves as the teacher model. This distillation process imposes constraints on the learnable prompt, preventing it from deviating too far during updates. Through self-text distillation, the process not only maintains low memory use but also reduces inference time during the testing phase. Additionally, SCP involves a fully test-time adaptation method, improving robustness directly during the testing phase without the need for retraining.

Overall, the contributions of our work can be summarized as:

- Our method employs conjugate pseudo-labels at test-time, using gradient information to effectively mitigate bias toward outlying samples while improving robustness to distribution shifts.
- Our method implements a novel self-text distillation strategy within the same model, acting as self-regularization to reduce overfitting.
- Our method is a fully test-time adaptation method that eliminates the need for retraining and directly improves the zero-shot generalization of CLIP by reducing both memory use and inference time.

## 2 Related Works

In this section, we present the study of vision-langauge models with prompt tuning and the methods of test-time adaptation.

### 2.1 Vision-Language Models with Prompt Tuning

Vision-Language Pre-trained Models [22, 33] (VLMs) serve as a bridge to unify the understanding of visual and textual data, achieving this through extensive pre-training that integrates both image and text inputs. Of these models, the CLIP model [29] stands out for its robust performance across a variety of tasks, including few-shot classification, zero-shot classification, and generalization tasks. CLIP is equipped with separate encoders for text and images and learns via self-supervised techniques using contrastive loss. It essentially aligns the representations of both textual and visual content, allowing the model to "understand" and interpret visual material in tandem with text descriptions. For this reason, CLIP boasts a great deal of utility in a range of settings and has proven to be quite effective.

One of the more recent developments in the realm of vision-language models is the application of prompt tuning. Prompt tuning emerged from the field of NLP as a means of effectively tailoring large-scale models for downstream tasks [7, 30]. This approach leverages textual prompts to direct the model's attention towards particular tasks, employing prompts as cues for the model to generate task-relevant responses. For instance, CLIP uses prompts such as *"a photo of a [object]"* to assess the match between an image and possible text descriptions, enhancing its classification capabilities based on the most fitting description.

Most early studies involved manually setting prompts. But finding appropriate prompts can be a time-consuming and challenging task. Hence, recent research has focused on enabling the model to learn more suitable prompts independently [18]. For example, the CoOp [50] method was the first approach to introduce learnable prompts for CLIP. The goal was to improve the model's ability to adapt to a task using a small set of training samples. While a significant advancement in theory, the actual mechanism often resulted in a model overfit for specific tasks. To address this, Co-CoOp [49] seeks to maintain CLIP's generalization performance across a range of tasks by applying meta-learning techniques that optimize the variation in prompts. By contrast, PLOT [2] employs optimal transportation to manage multiple prompts. Meanwhile, Maple [17] integrates learnable weights into both the image and text encoders, representing a step forward in enhancing CLIP's

learning capabilities. However, what all these methods have in common is that they require the CLIP model to be retrained. Plus, they only provide a modest boost to zero-shot generalization performance. Conversely, our approach with SCP is designed to directly enhance CLIP's zero-shot generalization potential during testing, eliminating the extensive retraining process. SCP focuses on exploiting the pre-trained model's inherent properties, enabling CLIP to effectively adapt to changing data.

## 2.2 Test-Time Adaptation

Test-time adaptation refers to updating the model during the testing phase [26, 37, 47], as distinct from traditional domain adaptation methods that require data from both the source and target domains [41, 45]. This approach does not need access to source data nor does it require retraining. Instead, it updates the model using information from the test data to mitigate the effects of distribution shift. Previous methods include entropy minimization, image augmentation, teacher-student model distillation, and the use of pseudo-labels. Entropy minimization [26, 37] aims to enhance the certainty of model predictions by minimizing the entropy of the test data, but it may lead to overfitting on specific high certainty test samples. Image augmentation techniques employ extensive augmentations to minimize marginal entropy [34, 47]. This approach seeks more stable results at the cost of an increased computational overhead and reduced inference speed, especially with larger-scale vision-language models. Teacher-student model distillation updates two models simultaneously to mitigate overfitting [8, 40]. But, again, this significantly increases the demand on computational memory. Pseudo-labeling [35, 39], which is the method SCP uses, involves updating the model with predictions as pseudo-labels. However, inaccurate pseudo-labels risk biasing the model toward outlying samples. Hence, to minimize this bias while reducing overfitting and maintaining efficiency, we adopted an innovative self-text distillation method. This approach, which only requires one model that performs both the teacher and the student roles, uses conjugate pseudo-labels to do the optimization based on both gradient and entropy information.

## 3 Preliminaries

In this section, we first introduce the structure of CLIP and the settings of the learnable prompt, followed by an explanation of the concepts of zero-shot generalization and test-time adaptation.

## 3.1 Introduction of CLIP

The CLIP model [29] consists of two encoders, one is an image encoder $f(\cdot)$, which is based on either ResNet[12] or ViT [36], and the other is a text encoder $g(\cdot)$, which is based on Transformer. CLIP pre-trains the contrast loss between image and text to get a better correspondence between image and text. To apply CLIP to specific downstream image classification tasks, a straightforward and effective method involves embedding text prompts, such as *"a photo of a"*, prior to the text encoder. These prompts, along with the [CLASS] token, are inputted to the text encoder. Consequently, the text encoder of CLIP is guided to generate a semantic representation that aligns with the corresponding input image. The inference

process of CLIP can be expressed as:

$$P(\hat{y} = i|\mathbf{x}) = \frac{\exp(\cos(\mathbf{w}_i, \mathbf{f}(\mathbf{x}))/\tau)}{\sum_{j=1}^{K} \exp(\cos(\mathbf{w}_j, \mathbf{f}(\mathbf{x}))/\tau)} \quad (1)$$

where $P(\hat{y} = i|\mathbf{x})$ represents the probability of the input $\mathbf{x}$ belonging to class $i$, $\cos(\mathbf{w}_i, \mathbf{f}(\mathbf{x}))$ is the cosine similarity between the class embedding $\mathbf{w}_i$ and the image feature $\mathbf{f}(\mathbf{x})$ from image encoder $f(\cdot)$, $\tau$ is a temperature parameter, and $K$ is the total number of classes.

The CoOp framework introduces a novel method that incorporates learnable prompts into the CLIP model while freezing the remaining parameters of the encoders [50]. This is achieved by formulating the prompts provided to the text encoder $g(\cdot)$ as follows:

$$\mathbf{t} = [\mathbf{V}_1][\mathbf{V}_2]\dots[\mathbf{V}_m]\dots[\mathbf{V}_M][\text{CLASS}] \quad (2)$$

where each $[\mathbf{V}_m]$, for $m = 1$ to $M$, represents a learnable vector with the same dimensionality as the word embeddings used in CLIP. The variable $M$ determines the number of contextual tokens within the prompt, and [CLASS] is the class name token.

For an image-text pair $(\mathbf{x}, \mathbf{t})$, the image feature $\mathbf{f}(\mathbf{x})$ is obtained from the image encoder $f(\cdot)$, and the text feature $\mathbf{g}(\mathbf{t})$ is obtained from the text encoder $g(\cdot)$. The prediction is then calculated by:

$$P(\hat{y} = i|\mathbf{x}) = \frac{\exp(\cos(\mathbf{g}(\mathbf{t}_i), \mathbf{f}(\mathbf{x}))/\tau)}{\sum_{j=1}^{K} \exp(\cos(\mathbf{g}(\mathbf{t}_j), \mathbf{f}(\mathbf{x}))/\tau)} \quad (3)$$

In this expression, for each individual prompt $\mathbf{t}_i$, the class token [CLASS] is substituted with the respective word embedding vector corresponding to the name of the $i$-th class.

## 3.2 Zero-shot Generalization

The concept of zero-shot generalization refers to the ability of a model to correctly categorize new, unseen data [10]. Specifically, let $\mathcal{X}^S = \{(\mathbf{x}_s, y_s)\}_{s=1}^{N_s}$ represent the dataset of seen classes, where $\mathbf{x}_s$ is a feature vector that can be obtained using a pre-trained deep learning model such as ResNet. The labels $y_s$ are the corresponding annotations for these feature vectors. $\mathcal{Y}^S = \{y_{s1}, y_{s2}, \dots, y_{sC_s}\}$ indicates the label set of seen classes within the label space $\mathcal{Y}$, where $C_s$ is the number of seen classes. Similarly, let $\mathcal{X}^T = \{(\mathbf{x}_t, y_t)\}_{t=1}^{N_t}$ represent the dataset of unseen classes, with $\mathcal{Y}^T = \{y_{t1}, y_{t2}, \dots, y_{tC_t}\}$ indicating the label set of unseen classes, where $C_t$ is the number of unseen classes. $\mathcal{Y} = \mathcal{Y}^S \cup \mathcal{Y}^T$ denotes the union of both the seen and unseen classes, ensuring $\mathcal{Y}^S \cap \mathcal{Y}^T = \emptyset$. The objective in zero-shot generalization is to train a model $f(\theta)$ on $\mathcal{X}^S$ and apply $f(\theta) : \mathcal{X}^T \rightarrow \mathcal{Y}^T$ to classify the test samples of unseen classes [28, 43].

## 3.3 Test-Time Adaptation

Test-time Adaptation (TTA) is the process of using test data information to update the model during test time. The aim is to adapt a pre-trained model to a target domain by adjusting the model parameters $\theta$ at test time [3, 26]. This process refines the model's generalization capability, improving performance on samples from different distributions. Fully test-time adaptation avoids the need to retrain the original model [20] and relies solely on using the test data for access to source data. It is particularly advantageous when access to source domain data is limited by privacy or storage considerations. TTA is achieved by minimizing an unsupervised

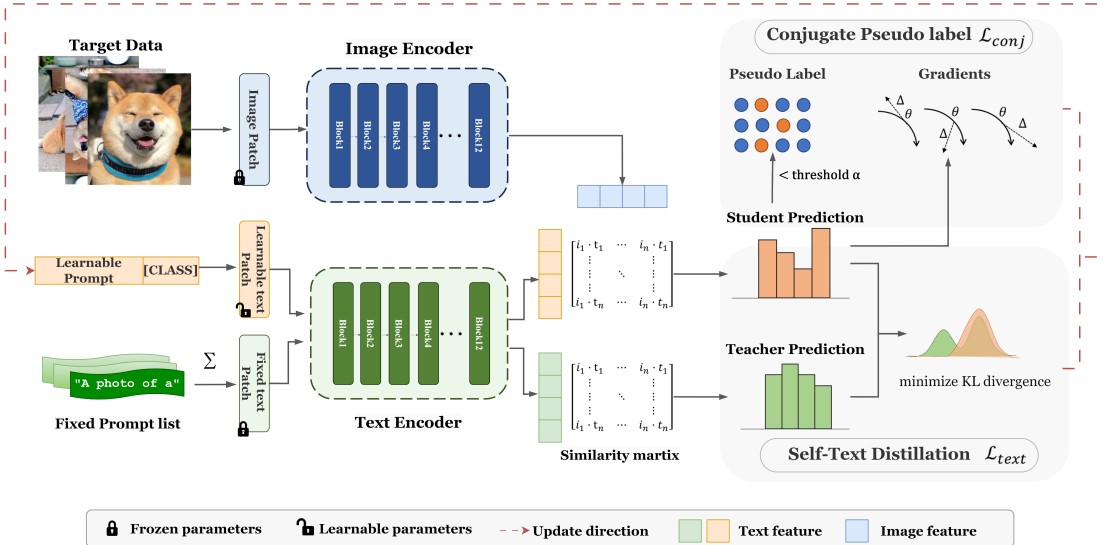

**Figure 1: Overview of the SCP framework.**

loss function $\mathcal{L}(\mathbf{x}_t, f(\theta))$ for test samples $\mathbf{x}_t \in \mathcal{X}^T$. The adaptation process is iterative, with each batch of test data leading to a parameter update through gradient descent:$\theta \leftarrow \theta - \alpha \nabla_\theta \mathcal{L}(\mathbf{x}_t, f(\theta))$, where $\alpha$ is the learning rate.

## 4 Methodology

In this section, we first provide an overview of the SCP framework. Then, the two core steps of the SCP algorithm are discussed – these being the conjugate pseudo-labelling process and self-text distillation.

### 4.1 Overview of SCP framework

Figure 1 provides an overview of the SCP method. SCP is based on a CLIP model, which extracts image and text features and uses these features to construct a similarity matrix that the student model then uses to make predictions. Next, the conjugate pseudo label loss is calculated based on the gradient information and the pseudo-labels generated from the entropy information of the predictions. Teacher predictions are made based on a fixed prompt list, while knowledge is distilled through KL divergence with the student predictions to result in the self-text distillation loss.

More specifically, the target data are divided into image patches, which are then input into the image encoder to obtain the image features. Then, learnable prompts generate learnable text patches, which are input into the text encoder to produce text features for the student model. These features are used to calculate the similarity matrix upon which the student predictions are based. The predictions from the student model with entropy values below a set threshold of $\alpha$ are then selected as the pseudo-labels. The loss is calculated from both the pseudo-labels and the gradient loss from the gradient information. As such, the overall conjugate pseudo label loss comprises the pseudo label loss and the gradient loss. This approach, which leverages the gradient information between the model's current output and the desired output, ensures that the

generated pseudo-labels will more effectively guide the model's learning process.

Concurrently, a fixed prompt list generates an assembled, fixed-text patch via the averaging method to generate the teacher model's prediction. Next, the student prediction and the teacher prediction are aligned using KL divergence, with the self-text distillation process aiming to minimize this divergence. This configuration helps the model learn stable knowledge representations from fixed prompts while exploring new or task-specific knowledge through adaptable prompts. Further, the process of self-regulation through text distillation maintains the model's regularity without modifications to the original CLIP architecture and without adding computational overhead.

In this approach, CLIP's image and text encoders are pre-trained, denoted as $f(\cdot)$ and $g(\cdot)$, respectively, and both are kept constant during the analysis. Given a specific target image $\mathbf{x}_t$, the corresponding image and text features are extracted via the learnable prompts described in Equation (2). This process is formalized as follows: $f(\mathbf{x}_t) = [i_1, i_2, \ldots, i_n]$ and $g(\mathbf{t}) = [t_1, t_2, \ldots, t_n]$. Then, the probability is computed as detailed in Equation (3).

The next two sections formally describe the process of selecting conjugate pseudo-labels and the self-text distillation in detail.

### 4.2 Conjugate Pseudo-labels

In Fully TTA situations where the model is updated only during the testing phase without access to source data, the accuracy of pseudo-labels is crucial for ensuring the effectiveness of the model. Previous pseudo-label methods only consider the prediction output from the model [24], which may enhance the original bias of the model, especially when the model is overconfident in its prediction.

Therefore, to enhance the reliability of pseudo-labels in fully TTA, we employ the conjugate pseudo-labels technique $y_t^{\text{conjugate}}$. This method considers both gradient information and entropy certainty. It initially involves entropy filtering to identify outputs with

lower entropy, leading to more credible predictions from the model. Mathematically, we define the entropy of the model's output probabilities $\mathbf{p}$ as:

$$H(\mathbf{p}) = -\sum_i p_i \log p_i \tag{4}$$

For each sample $\mathbf{x}_t$ in the target domain $\mathcal{X}^T$, we calculate the entropy $H(\mathbf{p}(\mathbf{x}_t))$ of the model's output. Samples with entropy lower than a threshold $\alpha$, i.e., $\{\mathbf{x}_t | H(\mathbf{p}(\mathbf{x}_t)) < \alpha\}$, are selected for generating more reliable pseudo-labels. The selected pseudo-labels $y_t^{\text{pseudo}}$ are then used to compute the cross-entropy loss for updating the model with the model's predictions $\hat{y}_{t,i}$:

$$\mathcal{L}_{\text{ce}} = -\sum_i y_t^{\text{pseudo}} \log \hat{y}_{t,i} \tag{5}$$

For the selected pseudo-label samples, we compute the conjugate pseudo-labels based on the gradient of a performance measure with respect to the model's parameters. In the CLIP model, the prediction is based on the cosine similarity between the image features and corresponding text features. The learnable parameters of the model include the prompt embeddings associated with the text input $\mathbf{t}$. Given a pair of image and text $(\mathbf{x}_t, \mathbf{t})$, the cosine similarity in the CLIP model is defined as:

$$\text{sim}(\mathbf{x}_t, \mathbf{t}) = \frac{\mathbf{f}(\mathbf{x}_t) \cdot \mathbf{g}(\mathbf{t})}{\|\mathbf{f}(\mathbf{x}_t)\| \|\mathbf{g}(\mathbf{t})\|} \tag{6}$$

where $\mathbf{f}(\mathbf{x}_t)$ represents the image features and $\mathbf{g}(\mathbf{t})$ represents the text features extracted by the respective encoders in the CLIP model. Consequently, the conjugate pseudo-label for an image $\mathbf{x}_t$ is determined by the gradient of the cosine similarity with respect to the image features $\mathbf{f}(\mathbf{x}_t)$:

$$y_t^{\text{conjugate}} = \nabla_{\mathbf{f}(\mathbf{x}_t)} \text{sim}(\mathbf{x}_t, \mathbf{t}) \tag{7}$$

This gradient provides direction for the model update, aiming to increase the cosine similarity for the correct image-text pairs, thereby improving the model's accuracy on the target task. The gradient of the cosine similarity with respect to the image features $\mathbf{f}(\mathbf{x}_t)$ is calculated as:

$$\nabla_{\mathbf{f}(\mathbf{x}_t)} \text{sim}(\mathbf{x}_t, \mathbf{t}) = \frac{\mathbf{g}(\mathbf{t})}{\|\mathbf{f}(\mathbf{x}_t)\| \|\mathbf{g}(\mathbf{t})\|} - \frac{\mathbf{f}(\mathbf{x}_t)(\mathbf{f}(\mathbf{x}_t) \cdot \mathbf{g}(\mathbf{t}))}{\|\mathbf{f}(\mathbf{x}_t)\|^3 \|\mathbf{g}(\mathbf{t})\|} \tag{8}$$

This takes both the alignment and magnitude of the image features into account, directing the updates to not only match the direction of the text features but also to adjust the scale of the image features.

Therefore, the conjugate loss is then:

$$\mathcal{L}_{\text{grad}} = -\sum_{i=1}^{C} y_t^{\text{pseudo}} \cdot \Big( z_i - \epsilon \cdot (1 - \text{sim}(\mathbf{x}_t, \mathbf{t})) \\ + \alpha \cdot y_t^{\text{conjugate}} \cdot (z_i - \text{sim}(\mathbf{x}_t, \mathbf{t})) \Big) \tag{9}$$

where $z_i$ are the logits. $\epsilon$ is a scaling factor to adjust the contribution of the similarity term in the loss function. $\alpha$ is a hyperparameter to control the influence of the conjugate gradient term in the loss.

For the total conjugate pseudo-label method, the parameters are updated using a combination of the cross-entropy loss and gradient loss:

$$\mathcal{L}_{\text{conj}} = \mathcal{L}_{\text{grad}} + \mathcal{L}_{\text{ce}} \tag{10}$$

By applying these steps, our goal is to utilize a more accurate pseudo-label to guide updates in the distribution shift scenario,

adapting the pre-trained model to the target domain more effectively.

## 4.3 Self-text Distillation

The conjugate pseudo-labels provide some additional update information from the gradient information, optimizing the robustness of the pseudo-labels at test-time. Furthermore, in addition to this process, a self-regularization method also constrains the optimization direction, further enhancing the robustness of the model's generalization capabilities. Existing test-time adaptation methods commonly employ prior knowledge and use two models in a teacher-student model distillation setup. However, for large-scale vision-language models, this approach significantly increases memory use and inference time. Minimizing these two overheads is often of critical practical importance in real-world applications.

Hence, SCP employs a novel self-text distillation strategy as a self-regularization mechanism. A fixed prompt list in CLIP's text encoder acts as the "teacher" and a learnable prompt mechanism in the same encoder acts as the "student". This method explicitly guides the training trajectory by imposing constraints, with the aim of maximizing mutual consistency between the prompt features and the frozen CLIP features. Ultimately, the goal is to mitigate the problem of overfitting. Formally, given a learnable prompt $\mathbf{t}$, and a set of fixed text templates $\{T_1, T_2, \ldots, T_n\}$, the text embedding process through the CLIP text encoder $g(\cdot)$ and the computation of fixed text feature embeddings $\mathbf{v}_{\text{fixed}}$ can be represented as follows:

$$\mathbf{v} = g(\mathbf{t}) \qquad \mathbf{v}_{\text{fixed}} = \frac{1}{n} \sum_{k=1}^{n} g(T_k) \tag{11}$$

The probability distributions $p(\cdot)$ and $q(\cdot)$, derived from the feature vectors $\mathbf{v}$ and $\mathbf{v}_{\text{fixed}}$ through a softmax function, are given:

$$p(i|\mathbf{v}) = \frac{\exp(\mathbf{v} \cdot \mathbf{w}_i)}{\sum_k \exp(\mathbf{v} \cdot \mathbf{w}_k)} \tag{12}$$

$$q(i|\mathbf{v}_{\text{fixed}}) = \frac{\exp(\mathbf{v}_{\text{fixed}} \cdot \mathbf{w}_i)}{\sum_k \exp(\mathbf{v}_{\text{fixed}} \cdot \mathbf{w}_k)} \tag{13}$$

where $\mathbf{w}_i$ represents the weight vector associated with the $i$-th category.

The self-text distillation loss, calculated using the KL divergence, is expressed as:

$$\mathcal{L}_{\text{text}} = \sum_i p(i|\mathbf{v}) \log \left( \frac{p(i|\mathbf{v})}{q(i|\mathbf{v}_{\text{fixed}})} \right) \tag{14}$$

The self-text distillation method focuses on aligning the learnable and fixed embeddings across the dataset. Consequently, the calculation of $\mathcal{L}_{\text{text}}$, which aims to average the losses, may need to be reconsidered based on the specific context of self-text distillation implementation. This approach not only conserves memory, which would otherwise be consumed by employing a teacher-student model simultaneously, but it also ensures more stable updates to the prompts.

Additionally, to enhance prediction stability and further reduce bias towards extreme samples, our methodology integrates a weighted averaging mechanism for updating the learnable prompts. This method accumulates the influence of past prompts, ensuring that

recent ones are adjusted in the context of their predecessors. The cumulative weighted prompt is computed as follows:

$$\mathbf{T} = \frac{\sum_{j=1}^{n} w_j \mathbf{t}_j}{\sum_{j=1}^{n} w_j}, \tag{15}$$

where the weight $w_j = e^{-(n-(j+1))}$ corresponds to the Gaussian-based weight for the $j$-th prompt in the sequence, $n$ is the total number of prompts, and $\mathbf{T}$ represents the updated prompt after applying the weighted average. In this scheme, the weights are configured to give greater influence to the more recent prompts, achieving a balance between embracing new updates and retaining the historical context, leading to a prompt that evolves smoothly over time and is both stable and robust.

In our model, the overall optimization method is:

$$\mathcal{L} = \lambda_1 \mathcal{L}_{\text{ce}} + \lambda_2 \mathcal{L}_{\text{grad}} + \lambda_3 \mathcal{L}_{\text{text}} \tag{16}$$

where, $\lambda$ is weight coefficient.

We provide the algorithm pseudo code in Appendix A.1.

## 5 Experiments

In this section, we first present the experimental settings. Then, we provide the results of a comparison on memory usage and the inference time. In addition, we present an ablation study, analyzing the impact of each component of the model.

### 5.1 Experiment settings

**Benchmarks.** We conduct zero-shot generalization evaluations on three benchmarks

- **Wild world distribution shift** [26] This evaluation is designed to assess the model's robustness to changes in wild-world distributions using 15 differently corrupted ImageNet variants. These are considered to be indistinguishable from real-world scenarios.
- **Cross-domain generalization** [29]. This evaluation assesses the generalization of the model to changes in fine-grained categorical datasets on 10 different categorical datasets.
- **Natural distribution shift** [34] This evaluation assesses the model's robustness to natural distribution shifts on four ImageNet variants. These data are considered out-of-distribution data for ImageNet.

**Baselines.** We compare the SCP with various test-time adaptation methods, all of which are based on comparisons using the CLIP model to ensure fairness. Each method qualifies as a full test-time adaptation approach, with no access to source data and no retraining involved and updates are only made at test time. We strictly adhered to the settings of these methods, reporting all results based on five runs. The comparison included:

- TENT [37], the first method to employ test-time entropy minimization.
- Pseudo-label [24], which updates the model using generated pseudo-labels.
- MEMO [47], the inaugural image augmentation method for test-time adaptation.
- CoTTA [40], which ustilizes a teacher-student model with image augmentation.

- RMT [8], a robust teacher-student model employing symmetric cross-entropy.
- SAR [26], a sharpness-aware and reliable entropy minimization method.
- TPT [34], the first test-time adaptation method integrating CLIP prompt tuning with minimization of marginal entropy through image enhancement.

**Datasets.** For the wild world distribution shift benchmark [26], we evaluate on the ImageNet-C dataset [15], which encompasses 15 types of image corruptions, including noise (Gaussian, shot, impulse), blur (defocus, glass, motion, zoom), weather (snow, frost, fog), and digital effects (brightness, contrast, elastic transformation, pixelation, JPEG compression). Subsequently, for the cross-domain generalization benchmark [29], we assess the performance on 10 distinct datasets. They include OxfordPets [27], StanfordCars [19], Caltech101 [9], DTD [4], EuroSAT [13], FGVCAircraft [23], Flowers102 [25], Food101 [1]. For the natural distribution shift benchmark [34], we evaluate on Imagenet [5], ImageNet-A [48], ImageNet-R [14], ImageNet-Sketch [38], and ImageNetV2 [31].

**Implementation Details.** We test baselines following the experimental setups and implementation of TENT, Pseudo Label, MEMO, CoTTA, RMT, SAR and TPT. For our method, SCP, we initiate with the 4-token prompt "a photo of a" and selected samples with entropy less than 1 for pseudo-label, setting $\lambda_1 = 10$, $\lambda_2 = 1000$, and $\lambda_3 = 5$. For cross-domain evaluations, we set the batch size to 1. For the other two benchmarks, the batch size is set to 100. The learning rate is 0.0025. The length of fixed prompt list is 10 prompts.

### 5.2 Results

In this section, we present the experimental results for the three benchmarks. Addtionally, we provide the evaluations of memory usage and inference time. For all benchmark tests, we compare each method separately based on the CLIP's ResNet50 and ViT-B/16 models. All methods require no retraining, with updates conducted only during testing.

**Wild world distribution shift.** In Table 1, we compare the performance of the various methods in 15 ImageNet corruption scenarios. When using the ResNet50 and ViT-B/16 models, the results from TENT and pseudo label suggest that relying solely on entropy minimization and pseudo-labels may reduce CLIP's generalization performance. Although TPT may show some improvement, it relies on image augmentation based on noise transformation, which may limit its effectiveness in corruption scenarios. However, our method, SCP, outperforms all other methods in terms of average accuracy and demonstrates improvements in almost all datasets. This is because SCP utilizes gradient information to obtain more accurate pseudo-labels and incorporates self-regulation to address distribution shifts.

**Cross-domain generalization.** In Table 2, we evaluate the generalization performance of various models across 10 cross-domain datasets. These datasets feature high-resolution images with a wide range of unseen categories, from animals to vehicles such as cars and airplanes. When using the ResNet50 and ViT-B/16 models, entropy minimization methods, such as TENT and SAR, become unstable. They solely rely on certainty, but the unseen categories

| Method | Gauss. | Shot | Impul. | Defoc. | Glass | Zoom | Motion | Snow | Frost | Fog | Bright. | Contr. | Elastic | Jpeg | Pixel. | Average |
|---|---|---|---|---|---|---|---|---|---|---|---|---|---|---|---|---|
| **CLIP-RN50** | 41.4 | 39.4 | 24.6 | 48.6 | 33.2 | 35.4 | 45.1 | 34.8 | 41.5 | 47.2 | 54.5 | 48.6 | 47.5 | 44.0 | 41.3 | 41.8 |
| • TENT | 32.5 | 38.5 | 15.8 | 49.8 | 23.2 | 35.3 | 45.0 | 31.9 | 34.0 | 48.3 | 55.9 | 49.8 | 48.4 | 38.5 | 41.3 | 39.2 |
| • Pseudo label | 41.5 | 39.6 | 24.6 | 48.8 | 33.3 | 35.5 | 45.2 | 34.0 | 41.7 | 47.4 | 54.8 | 48.8 | 47.8 | 44.3 | 41.5 | 41.9 |
| • MEMO | 40.6 | 38.4 | 20.5 | 47.1 | 30.8 | 33.3 | 42.6 | 33.2 | 37.5 | 45.5 | 53.6 | 47.1 | 45.4 | 40.1 | 40.1 | 39.7 |
| • CoTTA | 40.7 | 38.3 | 23.0 | 42.7 | 30.4 | 31.0 | 45.8 | 32.6 | 38.1 | 48.2 | 53.0 | 48.7 | 47.7 | 45.0 | 42.8 | 40.5 |
| • RMT | 39.1 | 39.1 | 18.4 | 47.3 | 29.7 | 34.4 | 42.2 | 33.6 | 40.8 | 46.1 | 53.4 | 47.3 | 36.3 | 44.9 | 40.7 | 39.6 |
| • SAR | 41.6 | 39.6 | **24.7** | 49.6 | 33.6 | 36.0 | 45.8 | 35.4 | 42.1 | 48.0 | 55.5 | 49.6 | 48.4 | 44.6 | 41.9 | 42.4 |
| • TPT | 42.1 | 40.3 | 24.0 | 41.9 | 34.5 | **36.0** | 46.3 | 34.8 | 41.9 | 48.6 | 55.9 | 50.4 | 49.5 | 46.6 | 43.8 | 42.5 |
| • SCP(Ours) | **42.8** | **40.7** | 22.4 | **50.7** | **35.3** | 31.1 | **48.0** | 34.8 | **43.9** | 49.4 | **57.6** | 50.7 | 50.5 | 47.7 | 43.9 | **43.3** |
| **CLIP-ViT-B/16** | 57.6 | 57.2 | 51.2 | 61.6 | 53.9 | 48.9 | 60.2 | 53.4 | 55.3 | 58.7 | 64.5 | 61.7 | 58.5 | 57.6 | 59 | 57.3 |
| • TENT | 58.6 | 58.2 | 51.7 | 41.2 | 54.7 | 49.2 | 61.1 | 54.5 | 56.2 | 34.1 | 65.7 | 42.1 | 59.3 | 59.1 | 60.1 | 53.7 |
| • Pseudo label | 58.0 | 57.4 | 51.4 | 61.9 | 54.2 | 49.1 | 60.5 | 53.9 | 55.5 | 58.9 | 64.8 | 61.9 | 58.9 | 58.0 | 59.4 | 57.6 |
| • MEMO | 58.5 | 57.0 | 50.2 | 60.0 | 52.2 | 47.8 | 58.6 | 52.8 | 55.7 | 58.0 | 63.4 | 61.0 | 58.3 | 59.0 | 59.9 | 56.8 |
| • CoTTA | 57.5 | 57.6 | 51.2 | 59.2 | 53.6 | 48.6 | 57.2 | 55.1 | 54.9 | 57.6 | 62.2 | 61.3 | 59.7 | 59.3 | 60.1 | 57.0 |
| • RMT | 58.4 | 57.7 | 51.5 | 60.7 | 54.0 | 44.5 | 60.5 | 55.9 | 53.4 | 56.2 | 63.0 | 62.2 | 58.2 | 57.2 | 59.0 | 56.8 |
| • SAR | 58.2 | 58.1 | 51.9 | 62.4 | 54.8 | 49.5 | 61.2 | 54.5 | 56.2 | 59.4 | 64.9 | 62.4 | 58.9 | 58.4 | 60.2 | 58.1 |
| • TPT | 57.3 | 57.1 | 51.7 | 56.5 | 54.5 | 49.4 | 60.5 | 55.2 | 56.8 | 59.0 | 65.3 | 62.3 | 59.8 | 58.8 | 60.1 | 57.6 |
| • SCP(Ours) | **59.6** | **59.4** | **53.6** | **63.9** | **56.5** | **50.9** | **62.5** | **56.5** | **58.1** | **60.9** | **67.2** | **63.9** | **60.8** | **60.2** | **61.9** | **59.7** |

**Table 1: Classification accuracy (%) for wild world distribution shift on ImageNet-C image classification task with the corruption severity level 1. Evaluation on ResNet50 and ViT-B/16. Bold number indicates the best result.**

| Method | Caltech | DTD | EuroSat | Craft | Food | Flower | Pets | Cars | Sun397 | Ucf101 | Average |
|---|---|---|---|---|---|---|---|---|---|---|---|
| **CLIP-RN50** | 83.2 | 39.7 | 26.6 | 14.5 | 76.0 | 63.7 | 83.2 | 55.5 | 60.2 | 59.4 | 56.2 |
| • TENT | 85.9 | 39.9 | 21.5 | 15.8 | 76.1 | 61.1 | 83.6 | 55.7 | 59.9 | 58.1 | 55.8 |
| • Pseudo label | 86.0 | 40.1 | 23.8 | 15.6 | 75.4 | 60.9 | 83.7 | 55.7 | 58.7 | 58.3 | 55.8 |
| • MEMO | 85.9 | 40.0 | 21.0 | 15.7 | 76.0 | 61.1 | 83.6 | 55.8 | 59.9 | 58.2 | 55.7 |
| • CoTTA | 86.8 | 41.8 | 23.9 | 15.9 | 76.3 | 59.4 | 83.0 | 54.4 | 55.3 | 58.0 | 55.5 |
| • RMT | 86.7 | 41.9 | 23.8 | 15.9 | 48.8 | 59.6 | 83.7 | 55.8 | 58.9 | 58.2 | 53.3 |
| • SAR | 86.0 | 39.7 | 23.9 | 13.4 | 76.0 | 61.1 | 83.6 | 55.8 | 58.6 | 58.3 | 55.6 |
| • TPT | 86.6 | **42.1** | 27.4 | 16.5 | 76.3 | 62.4 | 83.3 | 56.1 | 60.8 | 59.6 | 57.1 |
| • SCP(Ours) | **87.6** | 41.7 | **28.2** | **16.8** | **78.9** | **63.7** | **83.9** | **56.2** | **63.6** | **61.1** | **58.2** |
| **CLIP-ViT-B/16** | 93.3 | 44.2 | 42.0 | 23.6 | 83.6 | 67.4 | 88.2 | 65.4 | 62.5 | 65.1 | 63.5 |
| • TENT | 93.0 | 44.3 | 37.2 | 23.7 | 86.0 | 67.3 | 88.1 | 65.4 | 64.4 | 64.8 | 63.4 |
| • Pseudo label | 93.0 | 44.3 | 38.9 | 23.7 | 85.6 | 67.4 | 88.1 | 65.3 | 62.6 | 65.0 | 63.4 |
| • MEMO | 93.1 | 44.3 | 36.0 | 23.7 | 86 | 67.3 | 88.2 | 65.4 | 63.5 | 64.8 | 63.2 |
| • CoTTA | 92.9 | 46.2 | 46.8 | 24.4 | 86.5 | 65.9 | 86.9 | 65.7 | 63.5 | 65.0 | 64.4 |
| • RMT | 93.0 | 46.2 | 41.0 | 24.2 | 85.3 | 66.0 | 87.8 | 64.5 | 63.4 | 64.9 | 63.6 |
| • SAR | 93.0 | 44.6 | 41.0 | 13.1 | 85.8 | 67.6 | 88.2 | 65.2 | 62.5 | 64.8 | 62.6 |
| • TPT | **94.1** | **47.7** | 42.4 | 24.7 | 84.6 | 68.9 | 87.7 | **66.8** | 65.5 | **68.0** | 65.0 |
| • SCP(Ours) | 93.9 | 43.9 | **47.3** | **24.8** | **87.4** | **70.0** | **88.6** | 65.9 | **69.1** | 67.8 | **65.9** |

**Table 2: Classification accuracy (%) for cross-domain generalization on image classification task with 10 datasets. Evaluation on ResNet50 and ViT-B/16. Bold number indicates the best result.**

are too challenging to classify accurately. Teacher-student models like CoTTA and SAR exhibit stability in scenarios with noise, but they also struggle to predict new categories. The image augmentation method, TPT, shows some improvement but tends to overfit on the augmentation samples. Our method, SCP, has enhanced CLIP's performance on almost all datasets, particularly on more complex datasets like EuroSAT and SUN397, achieving an average accuracy improvement of 2% over CLIP. These results suggest that self-text distillation can mitigate overfitting and enhance stability when addressing cross-domain challenges.

**Natural distribution shift.** As shown in Table 3, for both ResNet50 and ViT-B/16 models, SCP consistently maintains a higher accuracy, with average accuracies of 46.5% and 61.4% respectively, which are

significantly above other competing methods. These results demonstrate SCP's superior generalization on a variety of challenging datasets.

**Memory usage and inference time.** We compare the performance of different methods in terms of memory usage, inference time, and accuracy, using the ViT-B/16 model as the benchmark. Entropy minimization methods like TENT and SAR use less memory but are less accurate. Teacher-student models, including RMT and CoTTA, consume more memory, with CoTTA also extending inference times, showcasing their computational complexity. TPT, utilizing image augmentation, exhibits the longest inference times, hinting at the added complexity from its image enhancement steps.

| Method | ImageNet | -A | -R | -Sketch | -V2 | Average |
|---|---|---|---|---|---|---|
| CLIP-RN50 | 58.2 | 21.8 | 56.1 | 33.3 | 51.4 | 44.2 |
| • TENT | 59.7 | 22.3 | 56.3 | 30.4 | 51.6 | 44.1 |
| • Pseudo label | 58.3 | 22.3 | 56.1 | 33.4 | 51.5 | 44.3 |
| • MEMO | 59.3 | 22.3 | 56.0 | 26.6 | 51.6 | 43.2 |
| • CoTTA | 58.8 | 22.1 | 54.7 | 30.5 | 50.5 | 43.3 |
| • RMT | 56.9 | 22.1 | 55.9 | 33.3 | 51.4 | 43.9 |
| • SAR | 58.9 | 22.3 | 56.3 | 33.6 | 51.5 | 44.5 |
| • TPT | **60.7** | 18.7 | 59.1 | 35.0 | **54.7** | 45.6 |
| • SCP(Ours) | 60.6 | **22.9** | **60.2** | **35.7** | 53.0 | **46.5** |
| CLIP-ViT-B/16 | 66.7 | 47.8 | 73.9 | 46 | 60.8 | 59.0 |
| • TENT | 67.6 | 48.7 | 74.6 | 46.9 | 61.0 | 59.8 |
| • Pseudo label | 66.9 | 48.6 | 74.1 | 46.5 | 60.8 | 59.4 |
| • MEMO | 65.8 | 48.6 | 74.4 | 22.1 | 60.8 | 54.3 |
| • CoTTA | 63.1 | 47.5 | 70.5 | 40.7 | 56.8 | 55.7 |
| • RMT | 63.5 | 47.7 | 74.0 | 46.1 | 60.6 | 58.4 |
| • SAR | 67.6 | 48.8 | 74.4 | 46.8 | 61.0 | 59.7 |
| • TPT | 68.4 | 47.8 | 77.0 | **47.0** | **63.0** | 60.6 |
| • SCP(Ours) | **68.8** | **50.5** | **78.7** | 46.5 | 62.6 | **61.4** |

**Table 3: Classification accuracy (%) for natural distribution shift on image classification task. Evaluation on ResNet50 and ViT-B/16. Bold number indicates the best result.**

SCP balances efficiency and accuracy, surpassing entropy minimization in speed and maintaining high accuracy, indicating effective management of distribution shifts with lower computational demands. For detailed results, please refer to the Appendix A.3.

## 5.3 Ablation study

In this section, we discuss the ablation study, focusing on the impact of self-text distillation and conjugate pseudo-labels, as well as the hyperparameters, including $\lambda 1, \lambda 2, \lambda 3$. In addition, we compare the experiments on prompt length, averaging methods, fixed prompt list length and threshold $\alpha$.

Table 4 presents the contribution of the self-text distillation ($\mathcal{L}_{\text{text}}$) and conjugate pseudo-labels ($\mathcal{L}_{\text{conj}}$) techniques toward enhancing the CLIP model's performance. As observed, applying conjugate pseudo-labels improves the average accuracy, which shows the robustness of using both the gradients and entropy to optimize the direction of the model updates. Next, integrating the self-text distillation further increases the average accuracy. This shows that consistency in the model is instrumental in boosting its generalization capabilities. Notably, in all scenarios, the combination of the two techniques yielded the highest average accuracy. Hence, we can conclude that this combined strategy does enhance the model's ability to generalize across diverse distribution shifts.

In the ablation study presented in Figure 2, we explore the impact of different hyperparameters on the performance of SCP. For $\lambda_2$ set to 100, we observe that an increase in $\lambda_1$ tends to initially improve accuracy but then shows a decrease, suggesting a sweet spot for the trade-off between these two loss components. As $\lambda_2$ increases to 1000, the pattern becomes less clear, potentially indicating a different dynamic at higher scales of gradient loss. Furthermore, varying $\lambda_3$, which controls the self-text distillation KL divergence, shows a non-monotonic relationship with accuracy.

| Method | Wild | Cross | Natural | Average |
|---|---|---|---|---|
| CLIP-RN50 | 41.8 | 56.2 | 44.2 | 47.4 |
| $+\mathcal{L}_{\text{conj}}$ | 41.7 | 56.7 | 46.2 | 48.2 |
| $+\mathcal{L}_{\text{text}}$ | 43 | 56.9 | 46.2 | 48.7 |
| $+\mathcal{L}_{\text{conj}} + \mathcal{L}_{\text{text}}$ | **43.3** | **58.2** | **46.5** | **49.3** |
| CLIP-ViT-B/16 | 57.3 | 64.5 | 59 | 60.3 |
| $+\mathcal{L}_{\text{conj}}$ | 58.9 | 63.8 | 60.7 | 61.1 |
| $+\mathcal{L}_{\text{text}}$ | 58.7 | 64.9 | 60.5 | 61.4 |
| $+\mathcal{L}_{\text{conj}} + \mathcal{L}_{\text{text}}$ | **59.7** | **65.9** | **61.4** | **62.3** |

**Table 4: Effects of the SCP components. Results are averaged over three benchmarks, evaluated on ResNet50 and ViT-B/16.**

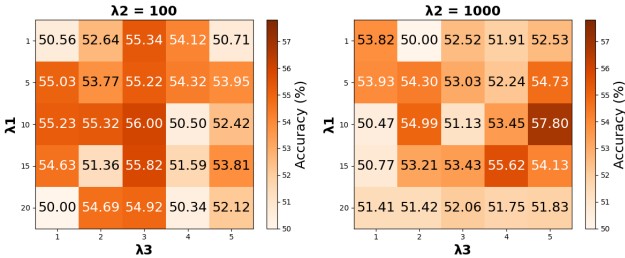

**Figure 2: Evaluation of SCP with selected hyperparameters on three benchmarks using the ViT-B/16 model. The numbers represent accuracy (%).**

Besides, for Prompt token length, a prompt length of 4 tokens provides the highest accuracy. Regarding Averaging methods, Gaussian weighting methods outperform both Exponential Moving Average (EMA) and Basic Moving Average (BMA). The threshold $\alpha$, equal to 1, provides the highest performance. For a more detailed results, please refer to Appendix A.2.

## 6 Conclusion and Future Work

In this study, we introduce a self-text distillation with conjugate pseudo-labels method (SCP) specifically designed for CLIP, aimed at enhancing its zero-shot generalization capability when dealing with changes in the distribution through test-time updates. SCP significantly enhances CLIP's zero-shot generalization performance across wild world distribution shifts, natural distribution shifts, and cross-domain generalization, while also accelerating inference speed and effectively mitigates the overfitting issues associated with previous test-time adaptation methods.

However, the current research focuses solely on individual datasets and a single type of distribution shift. In the future, we aim to investigate the impacts under continuous distribution changes and mixed distributions to further stabilize and robustify CLIP's zero-shot generalization.

## Acknowledgments

This work is supported by the Australian Research Council under Discovery Early Career Researcher Award DE220101075.

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
