# OpenReview forum: "Towards Robustness Prompt Tuning with Fully Test-Time Adaptation for CLIP's Zero-Shot Generalization"
_acmmm.org/ACMMM/2024/Conference — MM2024 Poster_

### Official Review · Reviewer_7JMc · 2024-05-24

**Rating:** 4
**Confidence:** 2

**Summary:**

This paper proposes a method to improve zero-shot CLIP’s zero-shot generalization, called Self-text distillation with Conjugate Pseudo-labels (SCP). SCP employs conjugate pseudo-label at test-time, using gradient information to reduce the bias on outlier samples. And SCP includes a self-text distillation strategy within the same model, to reduce over-fitting.

**Strengths:**

1， The paper writing is clear and easy to understand.

2， Using self-distillation to avoid memory burden is applicable.

3， The proposed method is a fully test-time adaptation method that can eliminate the need for retraining.

**Limitations:**

1，The improvement seems to be limited. Especially for dataset zoom, Impul, DTD etc.

2， How to choose threshold? Sensitivity analysis.

3, Why $L_{conj}$ brings negative impact on improvement for some dataset, as shown in Tab 4.

4, It is not clear to me SCP employs conjugate pseudo-label to reduce the bias on outlier samples.

**Suitability:**

3

---

### Official Review · Reviewer_WHP5 · 2024-05-25

**Rating:** 4
**Confidence:** 4

**Summary:**

The paper, "Towards Robustness Prompt Tuning with Fully Test-Time Adaptation for CLIP's Zero-Shot Generalization", presents a novel method called Self-Text Distillation with Conjugate Pseudo-labels (SCP). This method is designed to improve the zero-shot generalization capabilities of the CLIP (Contrastive Language-Image Pre-training) model by enhancing its robustness against distribution shifts and new categories encountered during test time. The SCP method employs conjugate pseudo-labels to mitigate bias toward outlying samples and uses self-text distillation to reduce overfitting. The key advantage of SCP is that it operates fully at test time, eliminating the need for retraining and ensuring minimal memory and inference time overheads.

**Strengths:**

1. SCP improves the zero-shot generalization performance without requiring retraining. This is highly beneficial in practical applications where retraining is not feasible due to time or computational constraints.
2. The paper includes a thorough evaluation across three benchmarks: wild world distribution shifts, cross-domain generalization, and natural distribution shifts. The results demonstrate that SCP outperforms existing state-of-the-art methods in both accuracy and inference speed.
3. SCP maintains low memory usage and inference time compared to other methods, making it practical for real-world applications.

**Limitations:**

1. The computational cost of calculating and usng gradient information for conjugate pseudo-labels might be non-trivial for larger models and datasets, potentially offsetting the benefits of reduced memory and inference time.
2. The method involves several hyperparameters, which may require fine-tuning for different tasks and datasets. This can be a tedious process and might limit the generalizability of the approach.

**Suitability:**

3

---

### Official Review · Reviewer_5htV · 2024-05-27

**Rating:** 4
**Confidence:** 2

**Summary:**

This paper addresses enhancing the zero-shot generalization capabilities of Vision-Language Models (VLMs), specifically the CLIP model, through a novel method called Self-Text Distillation with Conjugate Pseudo-labels (SCP). This method aims to improve robustness against distribution shifts at test time without retraining, focusing on reducing memory overhead and inference time.

**Strengths:**

- It seems that SCP outperforms existing state-of-the-art methods in zero-shot generalization scenarios across several benchmarks.
- Unlike other test-time adaptation techniques that increase inference time and memory usage, SCP is designed to be lightweight and fast

**Limitations:**

- It would be great if the authors could elaborate on potential trade-offs or limitations associated with using SCP.
- The method is dependent on CLIP, which may limit its applicability to other VLMs without substantial modifications. Can the principles of SCP be adapted for other types of neural network models or architectures beyond CLIP and VLMs?
- While SCP aims to reduce overfitting through self-regulation, the discussions on how it prevents overfitting are not sufficient to me.

**Suitability:**

3

---

### Meta-Review · Area_Chair_MtZM · 2024-07-04

**Recommendation:** Accept (Poster)
**Confidence:** 3

**Metareview:**

This submission focuses on improving the Contrastive Language-Image Pretraining (CLIP) model’s zero-shot generalization capabilities, which are challenged by new categories and distribution shifts. The proposed method, Self-Text Distillation with Conjugate Pseudo-labels (SCP), enhances CLIP’s robustness during test-time adaptation, outperforming existing methods without increasing memory overhead or inference time.

The initial comments are positive, with three borderline accepts. Reviewer 5htV requests clarification on SCP’s dependence on CLIP and addressing overfitting. Reviewer WHP5 raises concerns about computational cost and hyperparameters. Another reviewer asks for clarifications on sensitivity analysis and the mechanism of conjugate pseudo-labels. All reviewers acknowledge SCP’s simplicity and good performance.

After carefully reviewing the paper, comments, and rebuttal, AC finds that most concerns have been addressed, except for the computational cost of gradients and threshold choices. AC tends to give an acceptance recommendation. Please prepare for the camera-ready version carefully and ensure the suggested changes are incorporated into the revision. This recommendation has been discussed with SAC.